# Insomnia in Patients Undergoing Hypoglossal Nerve Stimulation Therapy for Obstructive Sleep Apnea

**DOI:** 10.3390/biology12010098

**Published:** 2023-01-09

**Authors:** Johannes Pordzik, Katharina Ludwig, Christopher Seifen, Tilman Huppertz, Katharina Bahr-Hamm, Christoph Matthias, Haralampos Gouveris

**Affiliations:** Department of Otolaryngology, Head and Neck Surgery, University Medical Center Mainz, Langenbeck Str. 1, 55131 Mainz, Germany

**Keywords:** obstructive sleep apnea, hypoglossal nerve, neurostimulation, insomnia, ISI (insomnia severity index), Epworth sleepiness scale (ESS), apnea/hypopnea index, oxygen desaturation, polysomnography, respiration

## Abstract

**Simple Summary:**

Obstructive sleep apnea (OSA) is a very prevalent disease. OSA is quite often accompanied by insomnia and the reported prevalence rate of comorbid insomnia and sleep apnea (COMISA) is 30–50%. Positive airway pressure (PAP) therapy delivered via nasal or oronasal interfaces has been established as the gold standard of therapy. Insomnia may complicate treatment of OSA and patients suffering from COMISA have reduced adherence to PAP-therapy. Respiration-coupled hypoglossal nerve stimulation (HGNS) has been established as a reliable therapeutic method in patients not tolerating PAP therapy. HGNS is associated with a significant improvement of sleep-related respiratory metrics, such as apnea-hypopnea index (AHI) and improvement of patient-reported sleep-related outcome measures. In this report, we provide evidence that HGNS therapy is also associated with a significant reduction of insomnia-related symptoms in OSA patients. Nonetheless, the preoperative severity of patient-reported insomnia symptoms is inversely correlated with the respiratory PSG-outcomes in OSA patients after HGNS therapy. In addition, we show a strong correlation between daytime sleepiness- and insomnia-related patient-reported outcomes. We suggest that insomnia should be carefully considered in future studies of excessive daytime sleepiness in OSA patients, especially those treated with HGNS.

**Abstract:**

Hypoglossal nerve stimulation (HGNS) is a treatment for obstructive sleep apnea (OSA) patients with intolerance of positive airway pressure therapy. Comorbid insomnia is quite prevalent in OSA patients. We investigated the impact of insomnia and excessive daytime sleepiness (EDS) on polysomnography metrics after HGNS treatment. Data of 27 consecutive patients (9 female; mean age 55.52 ± 8.6 years) were retrospectively evaluated. Insomnia was assessed using the ISI (insomnia severity index) and EDS using the Epworth sleepiness scale (ESS). The median ISI was reduced significantly 3 months after HGNS activation (preoperative: 19; postoperative: 14; *p* < 0.01). Significant correlations emerged between preoperative ISI and postoperative AHI (apnea/hypopnea index; Spearman’s rho = 0.4, *p* < 0.05), ∆ AHI (r = −0.51, *p* < 0.01) and ∆ ODI (oxygen desaturation index; r = −0.48, *p* < 0.05). ISI correlated strongly with EES both preoperatively (r = 0.46; *p* < 0.02) and postoperatively (r = 0.79; *p* < 0.001). Therefore, HGNS therapy is associated with a significant reduction of insomnia-related symptoms, in addition to the improvement in respiratory metrics in OSA. Nonetheless, the preoperative severity of patient-reported insomnia symptoms was inversely correlated with the respiratory PSG-outcomes after HGNS. Insomnia should be considered in studies of EDS in OSA patients, especially those treated with HGNS.

## 1. Introduction

Obstructive sleep apnea (OSA) is a very prevalent disease [1]. Recurrent episodes of partial and complete airway obstructions during sleep lead to apnea and hypopnea [2]. An increased risk of cardiovascular and metabolic conditions, such as coronary artery disease [3] and hypertension [4], hepatic steatosis [5] and stroke [6], among other comorbid conditions exists in OSA patients. The prevalence of OSA is on the increase [7]. OSA is quite often accompanied by insomnia and the reported prevalence rate of comorbid insomnia and sleep apnea (COMISA) is 30–50% [8]. The Insomnia Severity Index (ISI) is an established clinical tool for rating insomnia severity based on patient-reported outcomes [9]. ISI is a brief questionnaire that summarizes the subjective perception of insomnia symptoms, the consecutive distress and the impact on daily functioning.

Since OSA is a highly prevalent disease, over the years various options for OSA treatment have emerged such as positional sleep training therapy (i.e., induction or promotion of side sleeping using active or passive devices), mandibular advancement devices, continuous or auto-titrating positive airway pressure (PAP) therapy, weight loss and various surgical methods [10]. Continuous positive airway pressure (CPAP) therapy using nasal or combined oro-nasal delivery methods has been established as the gold standard of therapy in OSA [11]. Insomnia may complicate treatment of OSA [12] and patients suffering from COMISA have reduced adherence to PAP-therapy [8]. Even in patients suffering from OSA without comorbid insomnia, CPAP therapy is quite often not well tolerated [13].

Respiration-coupled hypoglossal nerve stimulation (HGNS) has been established as a reliable therapeutic method in patients not tolerating PAP therapy. By means of this therapy method through a submental cervical approach a cuff-based stimulating electrode is implanted to the patients under electrophysiological intraoperative guidance to selectively include only hypoglossal nerve branches that innervate the protruding and stiffening muscles of the tongue. The major tongue protrusor is the genioglossal muscle. At the same time, hypoglossal nerve branches that innervate the retractor muscles of the tongue (e.g., the styloglossal muscle) are excluded from the stimulating cuff electrode (Figure 1). Additionally, by means of an anterior thoracic approach at the level of the third rib an impulse generator providing the actual stimulation is secured on the pectoralis major muscle and a sensor lead, activated during each inspiratory phase of the breathing cycle is secured between the external and internal intercostal muscle at the level of the second intercostal space. The patient activates the device before going to sleep. As a result, depending on the level of activation, the tongue of the implanted OSA patient protrudes during the inspiratory phase of each breathing cycle during sleep.

HGNS is associated with a significant reduction of core sleep-related respiratory metrics, such as apnea-hypopnea index (AHI) and oxygen desaturation index (ODI), and of patient-reported outcome measures (PROMs), such as reduced subjective sleepiness as assessed using the Epworth Sleepiness Scale (ESS) [14]. However, the effect of comorbid insomnia on sleep-related metrics in patients with OSA treated by HGNS remains a matter of controversy. There are contradictory reports on HGNS use in COMISA patients; although one report showed an association of COMISA with reduced HGNS therapy compliance 12 months after implantation [15], other authors found no significant difference in HGNS use between patients with COMISA and those with OSA alone without comorbid insomnia [16]. To date no significant correlation between insomnia and objective PSG sleep metrics could be observed. The data for AHI, lowest ODI, and ESS were comparable between veterans with and without insomnia [16]. Additionally, no correlation between ISI values and the number of AHI or ODI events/h at the follow-up assessment after HGNS was shown [17]. Therefore, there is a need to find relevant predictors for postinterventional HGNS therapy success.

The aim of the present study was to evaluate the impact of HGNS therapy on insomnia-related symptoms as well as the association between patient-reported insomnia outcomes and objective PSG metrics. In addition, we investigated a possible association between patient-reported insomnia outcomes and patient-reported excessive daytime sleepiness (EDS) in OSA patients treated by HGNS therapy.

## 2. Materials and Methods

Patients who fulfilled the inclusion criteria and were implanted unilaterally with a respiration-coupled HGNS device (Inspire Medical Systems, Inc., Maple Grove, MN, USA) between February 2020 and June 2022 in our department were retrospectively evaluated. All patients included and evaluated in the framework of this study had fulfilled all the standard indication criteria for HGNS implantation as based on current guidelines [18]. These standard guideline criteria included an intolerance to PAP therapy, an apnea/hypopnea index (AHI) between 15 and 65 respiratory events (i.e., apneas and hypopneas) per hour sleep with less than 25% central apneas, as recorded on full-night diagnostic polysomnography (PSG), a body mass index (BMI) less than 35 kg/m^2^, an absence of a complete concentric collapse at the velar level during drug-induced sleep endoscopy (DISE) and an absence of chronic neurodegenerative or severe psychiatric disease.

Polysomnography was completed before and after HGNS implantation according to the American Academy of Sleep Medicine (AASM Inc., Darien, IL, USA) standard. Postoperative PSG was performed 95.63 ± 27.86 days after device activation. The PSG report of each individual was visually analyzed by sleep medicine experts for the following metrics: total number of apneic events per hour sleep (i.e., apnea index), total number of hypopneic events per hour sleep (i.e., hypopnea index), apnea/hypopnea index (AHI), total number of snoring events per hour sleep (i.e., snoring index), total number of oxygen desaturation events (≥4%) per hour sleep (oxygen desaturation index) and total number of cortical arousals per hour sleep (i.e., cortical arousal index).

Preoperatively all patients included had filled out both the ISI and ESS questionnaire. Postoperatively 23 out of the 27 patients had filled out both the ISI and ESS questionnaire.

The ISI is a brief seven-item questionnaire for assessment of the severity and impact of insomnia [9]. The questionnaire covers the different aspects of insomnia, mainly the difficulty falling asleep and staying asleep, the impairment of daily life due to sleep problems and sleep satisfaction in the last 2 weeks. Each item is scored from 0 = no problem to 4 = very severe problem, giving a total score ranging from 0 to 28. Based on the total score patients are categorized as having no clinically significant insomnia (0–7), subthreshold insomnia (8–14), clinical insomnia of moderate severity (15–21), or severe clinical insomnia (22–28).

The German version of the ESS was used to measure self-reported daytime sleepiness [19]. Eight items are scored on a four-point Likert scale that are summed to give a total score ranging from 0.0 to 24.0. Higher scores indicate greater daytime sleepiness.

### Statistical Analysis

Data were analyzed by use of SPSS 27 (IBM, Armonk, NY, USA). Categorical variables were provided as number and percentage (%), and continuous variables were described as mean ± standard deviation. Ordinal scaled variables were described as median. For comparisons between groups the Wilcoxon’s rank-sum test was performed. Spearman’s (rho-) correlation coefficients were calculated.

## 3. Results

The clinical and PSG data from 27 patients (9 female and 18 male) were assessed. Mean age at HGNS implantation was 55.52 ± 8.6 years and mean BMI was 30.03 ± 4.06 kg/m^2^.

A number of PSG-associated respiratory metrics were quite significantly improved after HGNS treatment (s. Table 1): AHI, apnea index, hypopnea index, snoring index, oxygen desaturation index (n/h) (*p* < 0.001, *p* < 0.01, *p* < 0.001 *p* < 0.001 and *p* < 0.05, respectively).

AHI in supine/other position did not differ significantly in both groups.

Two patients developed postoperative hematoma at the submental (stimulating electrode) cervical area within the first 7 days after implantation that resolved spontaneously. Two further patients developed an inflammatory reaction at the submental cervical implantation site that resolved under a 7-day course of oral (one patient) or 5-day course of intravenous (one patient) antibiotic therapy.

The median ISI score was significantly reduced postoperatively (preoperative ISI score = 19; postoperative = 14; *p* < 0.01). A significant reduction of the median ESS score was also shown postoperatively (preoperative ESS score = 14; postoperative = 9; *p* < 0.001).

Significant correlations (Spearman’s rho coefficient) were shown between the preoperative ISI score and postoperative AHI (r = 0.4, *p* < 0.05), change in AHI (∆ AHI; r = −0.51, *p* < 0.01), ∆ HI (r = −0.38, *p* < 0.05) as well as ∆ ODI (r = −0.48, *p* < 0.05) (s. Table 2 and Table 3 as well as Figure 2). The preoperative ESS score was inversely correlated with the postoperative arousal-index (r = −0.53; *p* < 0.01) (s. Table 3). Furthermore, the preoperative ISI-score correlated significantly with the preoperative ESS-score (r = 0.46; *p* < 0.02); additionally, a very strong and significant correlation between the postoperative ISI-score and the postoperative ESS-score was found (r = 0.79; *p* < 0.001).

## 4. Discussion

In this study, we demonstrate that both ISI-score as well as ESS-score were significantly reduced postoperatively in our cohort of HGNS-implanted OSA patients. We also provide evidence that the degree of insomnia measured by the ISI score preoperatively is significantly inversely correlated with the objective postoperative PSG metrics after HGNS therapy. More specifically, the preoperative ISI score is strongly and significantly associated with higher postoperative AHI as well as with lower degrees of AHI-, HI- and ODI-reduction. Patients with a high preoperative ESS score showed a particularly significant reduction of their postoperative arousal index on PSG.

With a high global prevalence (18–42%), comorbid insomnia and sleep apnea (COMISA) are the most common co-occurring sleep disorders [20]. So far, OSA and insomnia, even if occurring together, are treated separately with different treatment modalities. Continuous positive airway pressure (CPAP) therapy delivered via various nasal or oronasal interfaces has been established as standard therapy [11]. The standard therapy of insomnia is cognitive-behavioral therapy for insomnia [21]. However, PAP-therapy adherence rate was shown to be lower in patients with COMISA than in patients suffering from OSA alone [22]. Although a study showed no significant differences in HGNS adherence between patients with COMISA and those with OSA only [16], the trend to lower usage of therapy systems has often been demonstrated for HGNS therapy [15,17]. Furthermore, a study demonstrated worse scores and less improvement from baseline in patient-reported outcomes (ESS, FOSQ, overall satisfaction) in patients with moderate/severe insomnia at their follow-up assessment [17]. Despite these findings, a significant association between objective PSG metrics and insomnia was missing in the aforementioned study; this may be due to the fact that ISI score had not been evaluated preoperatively. On the contrary, we are not routinely using other questionnaires such as the Functional Outcomes of Sleep Questionnaire (FOSQ), which was used by Steffen et al. [17], and this may be a limitation in our study. Given that FOSQ may indeed provide additional patient-centered information on therapy effects, use of FOSQ may be considered in future studies on this topic. Nonetheless, our study is the first to show a significant reduction of insomnia-related symptoms parallel to the improvement of the respiratory PSG-metrics under HGNS-therapy. Based on this evidence, we suggest that HGNS may be a reliable therapy for both insomnia and OSA in individuals suffering from both conditions. In previous studies, a significant reduction of insomnia-related symptoms in OSA patients was reported under PAP-therapy [23]. Furthermore, to the best of our knowledge, we provide with this report for the first time, evidence for a significant inverse correlation between the ISI score and the HGNS therapy outcome.

ESS-score was significantly reduced after HGNS-therapy. This finding is in line with previous studies [17,24]. ISI-score and ESS-score correlated significantly both pre- and postoperatively. Higher insomnia symptoms and daytime sleepiness were shown for insomnia patients with severe fatigue [25]. Previously, one study could show that higher ISI values (greater insomnia severity) were associated with higher ESS scores in patients after HGNS-therapy [17]. Nonetheless, one should be cautious when interpreting ESS results, because ESS is a subjective questionnaire and this may represent a severe limitation regarding findings and conclusions in studies involving OSA patients [26]. This is a reason why more objective methods, based on automated EEG signal processing and analysis, are investigated to test for daytime sleepiness in OSA [27].

Furthermore, no significant correlation was observed for preoperative ISI score and postoperative cortical arousal index as well as the difference in cortical arousal index. Insomnia is often associated with hyperarousal processes from the molecular to the higher system level even if PSG sleep-abnormalities are minor reported [28]. Previously it could be demonstrated that HGNS therapy is able to significantly suppress cortical arousals [29]. However, higher ESS-scores were significantly associated with lower postoperative cortical arousal index. Frequent arousals often lead to excessive daytime sleepiness [30]. Previous studies, especially in OSA patients with compromised upper airway anatomy, have shown an association between ESS score and baseline (pre-interventional) arousal index on PSG [31].

There are many publications showing the efficacy and safety of HGNS therapy [14,32,33,34]. Furthermore, many publications showed an improvement of ESS under HGNS therapy [17,35,36]. However, our study is one of the first aiming to investigate any association between the preoperative ESS score and the post-operative HGNS therapy success. We provide evidence here that the preoperative ESS score has no significant correlation with the postoperative PSG-based results.

A limitation of our study is that it included a relatively small sample size and the data were acquired retrospectively. It should also be noted that the postoperative data presented here show the follow-up after 95.63 ± 27.86 days. However, the quality and completeness of the gathered data has been quite high. The relatively short-term follow–up (i.e., 3 months on average) may be considered another limitation of our study. As shown on Table 1, the median AHI after HGNS implantation was 25.4 events/h sleep.

Regarding the changes in median AHI after HGNS implantation, it is evident that in the short-term (namely, 3-month period after HGNS activation) the AHI reduction reported in our cohort, although statistically and clinically significant, does not fulfill the traditional Sher criteria for surgical treatment success as defined by Sher et al. 1996, namely a ≥50% AHI reduction to a final AHI ≤ 20/h [37]. More specifically, 26% of the patients reached the Sher success criteria after 3 months. Of note, our patients had a higher AHI and a higher BMI than previously reported cohorts [14,24,33]. This fact may indeed reduce effectiveness of HGNS treatment regarding the impact on AHI, at least in the short-term. Additionally, it may be that some features of the microstructure of cortical arousals, such as the cortical arousal intensity or cortical arousal entropy [38,39] in OSA patients, which are not depicted in the PSG macroscopic cortical arousal metric may further promote the hyperarousal state associated with insomnia in OSA and hence complicate OSA and OSA treatment. Nonetheless, we expect further improvement of the AHI with further continuation of the titration of the device’s neurostimulation parameters within the first 12 months after activation of the HGNS implant in our cohort.

Another issue that may influence post-operative AHI outcome in this particular group of patients may be patient adherence to the use of the HGNS device. OSA patients with comorbid insomnia may indeed have a reduced cumulative HGNS device usage time compared to OSA patients without insomnia, because they may simply spend more time awake in the night. This fact may compromise compliance. We did not check the HGNS device memory card of our patients for the cumulative device use to investigate compliance. This is a limitation of our study that should be seriously considered in future studies in this specific group of patients.

The process of finding the optimal HGNS stimulation level (within the available range of 0.1 to 5.0 V) may quite often last for months, with some patients needing even more than 12 months. Therefore, a further reduction of the AHI values 12 months after activation of the HGNS implant is expected. In case of residual AHI > 15 events/h after HGNS therapy, additional therapeutic measures such as weight loss, mandibular advance devices or sleep positional training may be suggested and tried, depending on the individual features of OSA disease and the compliance or preference of each separate OSA patient.

To date there are many studies showing the efficacy and safety of HGNS therapy [14,32,33,34]. Nonetheless, only a few studies have investigated the outcome of HGNS therapy in patients with co-occurring insomnia. These studies mainly focused on therapy adherence [15,16,17]. Data concerning predictors of HGNS therapy outcome are largely missing [40]. To the best of our knowledge, this is the first study investigating the outcome of HGNS therapy on insomnia-related symptoms as well as the association of ISI-score with the HGNS therapy success.

## 5. Conclusions

A significant reduction of insomnia-related symptoms results, in addition to a significant reduction of respiratory distress, in OSA patients after HGNS therapy. Furthermore, a significant inverse correlation between the degree of preoperative patient-reported insomnia, as measured by the ISI score, and the postoperative reduction of respiratory PSG metrics, such as AHI, HI and ODI, was found.

Moreover, a strong correlation between ESS and ISI scores, both pre- and post-operatively was shown. Therefore, insomnia should be carefully considered in future studies of both respiratory PSG-based outcomes as well as of excessive daytime sleepiness in OSA patients, especially those treated with HGNS. HGNS may provide a reliable therapy for both insomnia and OSA in individuals simultaneously affected by both conditions.

## Figures and Tables

**Figure 1 biology-12-00098-f001:**
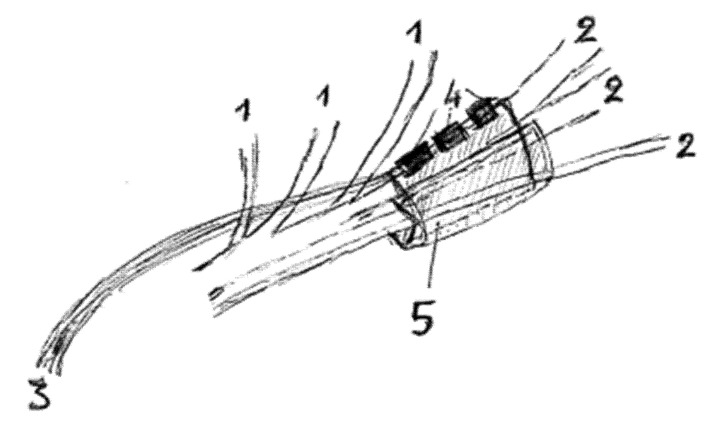
Drawing depicting the stimulating cuff electrode of the HGNS implant including the branches of the right-sided hypoglossal nerve, which innervated the protrusors and stiffeners of the tongue (1 = branches for the tongue retractor muscles-excluded from the cuff electrode), 2 = nerve branches for the protrusors and stiffeners of the tongue, 3 = cable connecting the stimulating cuff electrode to the impulse generator secured on the pectoralis major muscle, 4 = the three contacts of the electrode delivering the stimulation on the selected nerve branches, 5 = the cuff of the electrode).

**Figure 2 biology-12-00098-f002:**
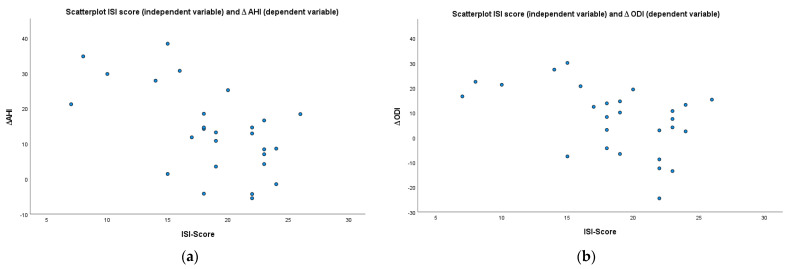
Scatterplot for the preoperative ISI score (independent variable) and (**a**) ∆ AHI and (**b**) ∆ ODI (dependent variable).

**Table 1 biology-12-00098-t001:** Respiratory PSG-based parameters pre- and postoperatively.

PSG-Metric	Preoperatively	Postoperatively	Comparison (*p*-Values)
AHI (n/h)	39.2 ± 12.93	25.45 ± 13.22	<0.001
AHI in supine position/AHI in other position	2.99 ± 1.38	3.57 ± 3.1	0.79
Apnea index (n/h)	15.14 ± 12.22	9.74 ± 11.05	<0.01
Hypopnea index (n/h)	24.04 ± 9.58	15.72 ± 8.36	<0.001
Snoring index (n/h)	269.99 ± 172.14	145.27 ± 168.53	<0.001
Oxygen desaturation index (n/h)	37.95 ± 15.2	30.59 ± 15.46	<0.05
Cortical arousal index (n/h)	20.53 ± 10.53	14.87 ± 8.63	0.11

**Table 2 biology-12-00098-t002:** Spearman’s rho (r) correlation coefficients between the preoperative ISI-score and ESS-score and the various postoperative PSG-metrics.

		AHI Post	AI Post	HI Post	ODI Post	Snoring Index Post	Arousal-Index Post
ISI score	r	0.4	0.32	0.19	0.29	0.11	−0.12
	*p*-value	<0.05	0.11	0.35	0.14	0.58	0.55
ESS score	r	0.06	0.16	−0.19	0.09	0.12	−0.53
	*p*-value	0.77	0.43	0.36	0.67	0.57	<0.01

**Table 3 biology-12-00098-t003:** Correlation coefficients between the preoperative ISI-score and ESS-score and the various differences (∆) in the pre- to post-operative PSG-metrics.

		∆ AHI	∆ AI	∆ HI	∆ ODI	∆ Snoring Index	∆ Arousal-Index
ISI score	r	−0.51	−0.29	−0.38	−0.48	−0.16	−0.32
	*p*-value	<0.01	0.15	<0.05	<0.05	0.42	0.1
ESS score	r	−0.17	−0.21	−0.07	−0.29	−0.22	0.23
	*p*-value	0.4	0.3	0.73	0.15	0.27	0.25

## Data Availability

Please contact the authors for data requests.

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
