# Peer review of "Insomnia in Patients Undergoing Hypoglossal Nerve Stimulation Therapy for Obstructive Sleep Apnea"

_biology, 2023, doi:10.3390/biology12010098_

Round 1

Reviewer 1 Report

Title : Insomnia in patients undergoing hypoglossal nerve stimulation therapy for obstructive sleep apnea.

This is an interesting study using the insomnia severity index to assess the impact of hypoglossal nerve stimulation. Other researchers can build on this foundational study . The text is well presented and concise. The statistical analysis is described, and the outcome put into perspective well.
There are just 2 small points I have for the authors.

Insomnia severity index and
1.    From what I gather the rationale for hypoglossal nerve stimulation is to causes the tongue to move to avoid obstructive nature while one sleeps. It would be nice for the reader to know this, so they don’t have to look it up.

2.    There is not much detail here about how the stimulator in placement along the nerve. For a reader’s sake it would be interesting to have some schematic to explain the set up
I had to search to get a visual  of the process, but I am not sure if it exactly what was done in this study  https://masseyeandear.org/treatments/

Author Response

We appreciate the time and effort that you dedicated to providing feedback on our manuscript and are very grateful for the insightful comments and valuable improvements to our submitted manuscript.

We have incorporated the suggestions made in our revised manuscript. All changes are highlighted in yellow.

Please find a point-by-point response to your valuable comments: 

Comment Nr. 1: We thank you for your suggestion. We have added a further description in the introduction , lines 66-87. "By means of this therapy method through a submental cervical approach a cuff-based stimulating electrode is implanted to the patients under electrophysiological intraoperative guidance to selectively include only hypoglossal nerve branches that innervate the protruding and stiffening muscles of the tongue. The major tongue protrusor is the genioglossal muscle. At the same time, hypoglossal nerve branches that innervate the retractor muscles of the tongue (e.g. the styloglossal muscle) are excluded from the stimulating cuff electrode (Figure 1). Additionally, by means of an anterior thoracic approach at the level of the third rib an impulse generator providing the actual stimulation is secured on the pectoralis major muscle and a sensor lead, activated during each inspiratory phase of the breathing cycle is secured between the external and internal intercostal muscle at the level of the second intercostal space. The patient activates the device before going to sleep. As a result, depending on the level of activation, the tongue of the implanted OSA patient protrudes during the inspiratory phase of each breathing cycle during sleep. "

Comment Nr. 2: we now added a drawing of ours depicting the stimulating cuff electrode in much more detail (Figure 1 of the revised manuscript)

The legend of the figure 1 describes with detail the respective structures:  "Figure 1. Drawing depicting the stimulating cuff electrode of the HGNS implant including the branches of the right – sided hypoglossal nerve which innervated the protrusors and and stiffeners of the tongue (1 = branches for the tongue retractor muscles-excluded from the cuff electrode), 2 = nerve branches for the protrusors and and stiffeners of the tongue, 3 = cable connecting the stimulating cuff electrode to the impulse generator secured on the pectoralis major muscle, 4 = the three contacts of the electrode delivering the stimulation on the selected nerve branches, 5 = the cuff of the electrode). "

Reviewer 2 Report

We have to congratulate the authors for a well-written  manuscript. There are several issues that concerned us.

1) Isi score should be included as a supplementary material.

2) Positional AHI should be included in the results pre and post treatment.

3) As a limitation this trial is not registered, please do it retrospectively or referred it as a limitation.

4) Complications and comorbidity of the surgery are not signed. Please refer to it.

5) As the final average result from AHI is 25, what was the final treatment for these patients? Keep with cpap, another surgery used MAD... If the follow-up was 3 months, do authors believe that these results will remain in the future.

6)Sher´s criteria are not complished about the results with hypoglossal pacemakers. Authors should consider this in the discussion, and why these results do not correspond with what it is published. 

7) There is severe limitation about Ess, as is a subjective questionnaire. Attached is Scharf´s manuscript. Authors should include this as a reference and point the  limitations on their use.

Author Response

Thank you very much for the time and effort that you dedicated to providing feedback on our manuscript. We are very grateful for the insightful comments and valuable improvements to our manuscript.

We have incorporated your and the other reviewers' suggestions  in our revised manuscript. All changes are highlighted in yellow in the revised manuscript.

We provide the following point-by -point response:

Comment Nr.1 :  Thank you for your comment. Since we are not absolutely sure if the ISI questionnaire is protected by copyright we had initially thought not to publish the questionnaire.

Comment Nr. 2:

Thank you. We have added a calculation for the ratio of AHI in supine position/AHI in non-supine position.

Please see line 167 as well as Table 1:

The ratio of AHI in supine/other position did not differ significantly in both groups.

Comment 3:  Data were analyzed retrospectively. It is mentioned in the method section. Furthermore, it has been added to the limitation part. Please see below:

Line 110-113:

Patients treated with unilateral implantation of the respiration-coupled HGNS (Inspire Medical Systems, Inc. Maple Grove, MN, USA) between February 2020 and June 2022 in our tertiary university medical center department of otorhinolaryngology were retrospectively evaluated.

Line 260-261:

A limitation of our study is that it included a relatively small sample size and the data were acquired retrospectively.

Comment 4: 

Complications and comorbidity of the surgery are added to the results section.

Please see Line 168-172:

Two patients developed postoperative haematoma at the submental (stimulating electrode) cervical area within the first 7 days after implantation that resolved spontaneously. Two further patients developed an inflammatory reaction at the submental cervical implantation site that resolved under a 7-day course of oral (one patient) or 5-day course of intravenous (one patient) antibiotic therapy. 

 Comment Nr. 5:  Thank you for this comment. We expect further improvement of the AHI with further continuation of the titration of the device’s neurostimulation parameters within the first 12 months after activation of the HGNS implant in our cohort.

Please see line 278-280:

 Nonetheless, we expect further improvement of the AHI with further continuation of the titration of the device’s neurostimulation parameters within the first 12 months after activation of the HGNS implant in our cohort.

Comment Nr. 6: 

Thank you! Sher`s criteria has been added and discussed in the discussion section.

Please see line 266-280:

Regarding the changes in mean AHI after HGNS implantation, it is evident that in the short-term (namely, 3-month period after HGNS activation) the AHI reduction reported in our cohort, although statistically and clinically significant, does not fulfill the traditional Sher criteria for surgical treatment success as defined by Sher et al. 1996, namely a ≥50% AHI reduction to a final AHI ≤ 20/h [37]. More specifically, 26% of the patients reached the Sher success criteria after 3 months. Of note, our patients had a higher AHI and higher BMI than previously reported cohorts [14, 24, 33]. This fact may indeed reduce effectiveness of HGNS treatment regarding the impact on AHI, at least in the short-term. Additionally, it may be that some features of the microstructure of cortical arousals, such as the cortical arousal intensity or cortical arousal entropy [38, 39] in OSA patients, which are not depicted in the PSG macroscopic cortical arousal metric may further promote the hyperarousal state associated with insomnia in OSA and hence complicate OSA and OSA treatment. Nonetheless, we expect further improvement of the AHI with further continuation of the titration of the device’s neurostimulation parameters within the first 12 months after activation of the HGNS implant in our cohort.

Please see line 289-295:

The process of finding the optimal HGNS stimulation level (within the available range of 0,1 to 5,0 Volt) may quite often last for months, with some patients needing even more than 12 months. Therefore, a further reduction of the AHI values 12 months after activation of the HGNS implant is expected. In case of residual AHI > 15 events /h after HGNS therapy, additional therapeutic measures such as weight loss, mandibular advance devices or sleep positional training may be suggested and tried, depending on the individual features of OSA disease and the compliance or preference of each separate OSA patient.  

Comment Nr. 7: 

This reference has been added and discussed in the discussion part.

Please see line 239-243:

Nonetheless, one should be cautious when interpreting ESS results, because ESS is a subjective questionnaire and this may represent a severe limitation regarding findings and conclusions in studies involving OSA patients [26]. This is a reason why more objective methods, based on automated EEG signal processing and analysis, are investigated to test for daytime sleepiness in OSA [27].

Reviewer 3 Report

This is a well written manuscript on the effect of HGNS on patients withcomorbid insomnia and sleep apnea (COMISA). It is the first comparing pre and post findingsand thus deserves publication. However, it requires some minor improvements: 1.- Why the authors did not use other questionaries such as Functional Outcomes of SleepQuestionnaire (FOSQ), as done by Steffen et al. 2.- How the authors explain the low rate of improvement of AHI? Although significant, it is below Sher's classical criteria. 3.- Have the authors analyzed adherence to HGNS in patients with or without
insomnia? 4.- With such a short 3-month follow-up, I suggest the authors to wait for a 12 month follow-up to see if results improve.

Author Response

We appreciate the time and effort that you dedicated to providing feedback on our manuscript and are very grateful for the insightful comments and valuable improvements to our submitted manuscript.

We have incorporated the suggestions made by the other reviewers and you in our revised manuscript. All changes are highlighted in yellow. 

Please consider our point-by -point responses:

Comment Nr. 1: 

Thank you. We have mentioned it as a limitation in the discussion section.

Please see line 222-226:

On the contrary, we are not using routinely other questionnaires such as Functional Outcomes of Sleep Questionnaire (FOSQ), as done by Steffen et al. [17] and this may be a limitation in our study. Given that FOSQ may indeed provide additional patient-centered information on therapy effects, use of FOSQ may be considered in future studies on this topic.

Comment Nr. 2:

Thank you! Sher`s criteria has been added and discussed in the discussion part.

Please see line 266-280:

Regarding the changes in mean AHI after HGNS implantation, it is evident that in the short-term (namely, 3-month period after HGNS activation) the AHI reduction reported in our cohort, although statistically and clinically significant, does not fulfil the traditional Sher criteria for surgical treatment success as defined by Sher et al. 1996, namely a ≥50% AHI reduction to a final AHI ≤ 20/h [37]. More specifically, 26% of the patients reached the Sher success criteria after 3 months. Of note, our patients had a higher AHI and higher BMI than previously reported cohorts [14, 24, 33]. This fact may indeed reduce effectiveness of HGNS treatment regarding the impact on AHI, at least in the short-term. Additionally, it may be that some features of the microstructure of cortical arousals, such as the cortical arousal intensity or cortical arousal entropy [38, 39] in OSA patients, which are not depicted in the PSG macroscopic cortical arousal metric may further promote the hyperarousal state associated with insomnia in OSA and hence complicate OSA and OSA treatment. Nonetheless, we expect further improvement of the AHI with further continuation of the titration of the device’s neurostimulation parameters within the first 12 months after activation of the HGNS implant in our cohort.

Comment Nr. 3:  

Adherence to HGNS in patients with or without insomnia has not been analyzed. This is a limitation of our study, that has been added to the discussion part.

Please see line 281-288:

Another issue that may influence post-operative AHI outcome in this particular group of patients may be patient adherence to the use of the HGNS device. OSA patients with comorbid insomnia may indeed have a reduced cumulative HGNS device usage time compared to OSA patients without insomnia, because they may simply spend more time awake in the night. This fact may compromise compliance. We did not check the HGNS device memory card of our patients for the cumulative device use to investigate compliance. This is a limitation of our study that should be seriously considered in future studies in this specific group of patients.

Comment Nr. 4: 

Thank you for this comment and your expert suggestion. We have added this comment in the “Discussion”.

Please see Lines 289-292:

The process of finding the optimal HGNS stimulation level (within the available range of 0,1 to 5,0 Volt) may quite often last for months, with some patients needing even more than 12 months. Therefore, a further reduction of the AHI values 12 months after activation of the HGNS implant is expected.

Round 2

Reviewer 2 Report

Authors have answered properly all suggestions asked by this reviewer.